# Influence of Graft Positioning during the Latarjet Procedure on Shoulder Stability and Articular Contact Pressure: Computational Analysis of the Bone Block Effect

**DOI:** 10.3390/biology11121783

**Published:** 2022-12-08

**Authors:** Rita Martins, Carlos Quental, João Folgado, Ana Catarina Ângelo, Clara de Campos Azevedo

**Affiliations:** 1IDMEC, Instituto Superior Técnico, Universidade de Lisboa, Av. Rovisco Pais, 1049-001 Lisboa, Portugal; 2Hospital CUF Tejo, Av. 24 de Julho, 1350-352 Lisboa, Portugal; 3Hospital dos SAMS de Lisboa, Rua Cidade de Gabela, 1849-017 Lisboa, Portugal

**Keywords:** glenohumeral instability, bony defect, Latarjet, coracoid graft, finite element method

## Abstract

**Simple Summary:**

By osteotomizing the coracoid process from the scapula and transferring it, along with the conjoint tendon, to the anteroinferior glenoid rim, the Latarjet procedure treats recurrent anterior glenohumeral (GH) instability in patients with large anterior glenoid bone defects. One key factor affecting its efficacy is the positioning of the graft on the glenoid rim, which impacts not only GH joint stability, through the so-called bone block effect, but also the joint contact mechanics; however, limited data exist on the best graft placement. In this study, finite element models that consider different medial–lateral graft placements for the Latarjet procedure were applied to evaluate GH joint stability and articular contact pressures. The results showed that the contribution of the bone block effect to GH joint stability increased with bone graft lateralization; however, the lateralization of the bone graft was also associated with an increase in peak contact pressures, which raises concerns regarding the risk of osteoarthritis. A trade-off seems to exist between GH joint stability provided by the bone block effect and the risk of osteoarthritis. This study provides valuable information regarding the influence of bone graft placement on the bone block effect of the Latarjet procedure.

**Abstract:**

The Latarjet procedure is the most popular surgical procedure to treat anterior glenohumeral (GH) instability in the presence of large anterior glenoid bone defects. Even though the placement of the bone graft has a considerable influence on its efficacy, no clear indications exist for the best graft position. The aim of this study was to investigate the influence of the medial–lateral positioning of the bone graft on the contact mechanics and GH stability due to the bone block effect. Four finite element (FE) models of a GH joint, with a 20% glenoid bone defect, treated by the Latarjet procedure were developed. The FE models differed in the medial–lateral positioning of the bone graft, ranging from a flush position to a 4.5 mm lateral position with respect to the flush position. All graft placement options were evaluated for two separate shoulder positions. Anterior GH instability was simulated by translating the humeral head in the anterior direction, under a permanent compressive force, until the peak translation force was reached. Joint stability was computed as the ratio between the shear and the compressive components of the force. The lateralization of the bone graft increased GH stability due to the bone block effect after a 3 mm lateralization with respect to the flush position. The increase in GH stability was associated with a concerning increase in peak contact pressure due to the incongruous contact between the articulating surfaces. The sensitivity of the contact pressures to the medial–lateral positioning of the bone graft suggests a trade-off between GH stability due to the bone block effect and the risk of osteoarthritis, especially considering that an accurate and consistent placement of the bone graft is difficult in vivo.

## 1. Introduction

The glenohumeral (GH) joint is the most often dislocated joint in the human body, with most dislocations occurring in the anterior direction [1,2]. About 50% of GH dislocations occur in a young and active population, which translates into a significant societal impact due to productivity losses and an overburdening of healthcare providers [3]. Anterior GH dislocation causes structural damage to the bony and soft tissues, which further increases instability and leads to recurrence. Recurrence rates of anterior GH instability as high as 87% have been reported in the literature [4].

In young and active patients, surgical treatment produces lower recurrence rates, better range of motion restoration, and a quicker return to daily-living activities than conservative treatment [2,4,5]. The Latarjet procedure is the most popular method for the treatment of recurrent anterior GH instability with large anterior glenoid bone defects. One key factor affecting the outcome of the Latarjet procedure is the positioning of the graft on the glenoid rim, especially in the medial—lateral direction. Grafts placed too medially are associated with high recurrence rates of GH instability [6], whereas grafts placed too laterally are associated with an increased risk of osteoarthritis because the overhang of the graft is likely to damage the humeral head cartilage [6,7]. Considering that glenoid arc and depth reconstruction is essential for joint stability, the graft is recommended to be placed flush to the glenoid surface [8]; however, tolerances with differences as large as 5 mm are accepted in the literature as proper placements [9,10].

As small changes in graft positioning can greatly influence the efficacy of the Latarjet procedure, the purpose of this study was to investigate how the medial–lateral positioning of the graft impacted contact mechanics and GH stability through the bone block effect, using three-dimensional finite element models. The hypotheses were that graft lateralization would be required for the bone block effect to contribute to GH stability, and that both GH stability and peak contact pressure on the humeral head cartilage would increase with graft lateralization.

## 2. Materials and Methods

### 2.1. Geometric Model

A three-dimensional (3D) geometric model of the right GH joint, including the scapula, humerus, and glenoid labrum, was built from the medical image data from the Visible Human Project [11]. For the sake of computational simplicity, the humerus was cut at its surgical neck. The articular cartilages, covering the glenoid and humeral head, were modeled in SolidWorks (Dassault Systèmes, Waltham, MA, USA) following published anatomical data. For the glenoid cartilage, a 2 mm extrusion of the glenoid surface was performed along the lateral direction and a Boolean cut operation was applied between the glenoid cartilage and labrum to ensure consistency between them [12,13]; for the humeral head cartilage, a 1 mm outward offset of the humeral articulation surface was performed [14,15], and the space in between was filled to create a solid structure.

Since the Latarjet procedure is normally recommended in the presence of large glenoid bone defects [16], a glenoid bone defect comprising 20% of the glenoid’s width was simulated, as shown in Figure 1a. The procedure was as follows: an osteotomy line was drawn parallel to the long axis of the glenoid, based on Yamamoto et al. [17], and the scapula, glenoid cartilage, and labrum were cut using the plane perpendicular to the glenoid surface and containing the osteotomy line. For the simulation of the Latarjet procedure (Figure 1b), the distal part of the coracoid process was osteotomized, the inferior surface of the bone graft was flattened, and the graft was positioned flush with the articular surface of the glenoid according to the instructions of an orthopedic shoulder surgeon (C.d.C.A.). This position is hereafter referred to as “ML-0 mm”. Two 30 mm long, full-threaded, 3.5 mm cortex screws, based on the Universal Small Fragment System of DePuy Synthes, were used to fix the bone graft. These were inserted perpendicular to the glenoid bone defect surface, centered on the graft surface, and their head centers were placed at a distance of 7.5 mm.

To evaluate the influence of the medial—lateral positioning of the graft on the outcome of the Latarjet procedure, three additional configurations (in addition to the ML-0 mm position) were considered by laterally translating the graft 1.5 mm, 3.0 mm, and 4.5 mm from the ML-0 mm position (Figure 2). These positions are hereafter referred to as “ML-1.5 mm”, “ML-3.0 mm”, and “ML-4.5 mm”, respectively. Medial translations of the bone graft from the ML-0 mm position were disregarded because their outcome was assumed to be similar to that of the ML-0 mm position regarding the bone block effect. For the sake of comparison with the literature, Table 1 quantifies the positions of the bone graft according to the axial circle method, described by Kany et al. [18].

All graft positions were evaluated for two shoulder positions, which were selected based on previous biomechanical studies [17,19,20]. These were: a 0° abduction in the scapular plane in neutral rotation, and a 60° abduction in the scapular plane with 45° of external rotation. Abduction angles are defined relative to the scapula, i.e., they are GH joint angles. These positions were simulated by rotating the humeral head around its geometric center.

### 2.2. Finite Element Models

Finite element models of the geometric models of the shoulder were developed in Abaqus (Dassault Systèmes, Waltham, MA, USA). For the sake of computational simplicity, the humerus was assumed rigid [21]. The scapula was assigned inhomogeneous isotropic linear elastic material properties taking into consideration the relationship between Young’s modulus *E* and bone density ρ [22]:(1)$$E\left(\mathrm{MPa}\right)=\left\{\begin{array}{l} 1049.45\rho^{2}\\ 3000\rho^{3} \end{array}\right.\;if\;\begin{array}{r} \rho \leq 0.35g/{\mathrm{cm}}^{3}\\ 0.35<\rho \leq 1.8\;g/{\mathrm{cm}}^{3} \end{array}.$$

Bone densities were estimated using CT data that assumed a linear relationship with Hounsfield Units and were obtained using the Abaqus plug-in “bonemapy” [23,24]. The glenoid labrum and articular cartilages were modeled as hyperelastic Neo-Hookean materials. The glenoid labrum was assigned a Young’s modulus of 46.6 MPa and a Poisson’s ratio of 0.4 [25], and the articular cartilages were assigned a Young’s modulus of 12.5 MPa and a Poisson’s ratio of 0.49 [26]. The screws were modeled as linearly elastic materials with a Young’s modulus of 113.8 GPa and a Poisson’s ratio of 0.3 [8].

Interactions between bone–cartilage, bone–labrum, labrum–cartilage (of the glenoid), and bone–graft were assumed to be bonded, while interactions between cartilage–cartilage, labrum–cartilage (of the humeral head), and graft–cartilage (of the humeral head) were assumed to be under frictionless contact [8,14].

Like in previous studies [17,27], anterior GH instability was simulated by translating the humeral head in the anterior direction, under a permanent compressive force, until the peak translational force (anterior translation shear force) was reached. Different simulations were performed with compressive forces of 50 N and 100 N. The rotations of the humerus were constrained to keep the same arm position throughout the analysis, and nodes in the vicinity of the *trigonum spinae* and *angulus inferior* bony landmarks of the scapula were fixed in all directions to avoid rigid body motion [28], as illustrated in Figure 3.

Due to the complex geometry of the structures, tetrahedral (C3D4) elements were used to create the finite element meshes. Hybrid (C3D4H) elements were considered for the articular cartilages. Mesh density was defined based on sensitivity analyses in which the peak translational force was evaluated for the healthy glenohumeral joint model and the peak contact pressure was evaluated for the glenohumeral joint model with a 20% glenoid bone defect. Mesh refinement, defined from average element sizes of 0.8 mm to 0.2 mm, with 0.1 mm increments, was stopped when results changed by less than 5% in two consecutive iterations [29,30]. The total number of elements and nodes resulting from the selected mesh density ranged from 1,775,000 to 1,840,000 and from 350,000 to 390,000, respectively, in all Latarjet finite element models.

### 2.3. Validation of Finite Element Models

Before studying the Latarjet procedure, the finite element models of the healthy GH joint and the GH joint with a glenoid bone defect were validated using experimental data on anterior GH instability, which provided confidence in the followed methodology [17,31]. Modeling conditions reproduced the experimental tests. Following the study of Lippit et al. [31], the humeral head in the healthy GH joint model was compressed onto the glenoid by a 50 N force and was translated in the anteroinferior, anterior, and anterosuperior directions. Joint stability was evaluated in each direction by computing the ratio between the shear and the compressive components of the GH joint reaction force, given by:(2)$$Stability\;ratio\;\left(SR\right)=\frac{Peak\;translational\;force}{Compressive\;force}.$$

A similar procedure was followed for the finite element model of the GH joint with a 20% glenoid bone defect, but joint stability, given by *SR*, was only evaluated along the anterior direction, following the study of Yamamoto et al. [17].

### 2.4. Analysis of the Medial–Lateral Bone Graft Positions

To compare the different medial–lateral graft positions, joint stability and contact pressures on the humeral head cartilage were evaluated. The stability of the GH joint was computed by *SR*, expressed in Equation (2). For the sake of comparison, GH contact pressures were evaluated from the beginning of the humeral head translation until the peak translational force was reached.

## 3. Results

Figure 4 presents the stability ratios obtained for the healthy GH joint and the GH joint with a 20% glenoid bone defect, along with the experimental data available in the literature. The glenoid bone defect decreased shoulder stability.

The bone block effect of the Latarjet procedure contributed to GH stability only in the ML-3.0 mm and ML-4.5 mm positions, regardless of the shoulder position, as shown in Figure 5. For the ML-4.5 mm position, GH stability was greater than in the healthy GH joint. No differences were observed between the ML-0 mm and ML-1.5 mm positions. From a qualitative point of view, the impact of the bone block effect of the Latarjet procedure on GH stability was similar for all the shoulder positions studied.

The evolution of the peak contact pressures with anterior translation of the humeral head is presented in Figure 6. Contact pressures in the ML-0 mm and ML-1.5 mm positions were always below the failure stress of the articular cartilage, which was assumed to be 29.5 MPa based on Riemenschneider et al. [32]. For these positions, no contact occurred between the humeral head cartilage and the bone graft. Both the ML-3.0 mm and ML-4.5 mm positions showed peak contact pressures above 29.5 MPa. However, while these much higher peak contact pressures were only reached after some anterior translation of the humeral head for the ML-3.0 mm position, for the ML-4.5 mm position they were reached before the humeral head was even anteriorly translated. During the compression of the humeral head against the glenoid cavity, it contacted the bone graft, as depicted in the Appendix A, which increased the peak contact pressure.

## 4. Discussion

Using 3D finite element analyses, this study investigated how the medial–lateral positioning of the graft during the Latarjet procedure impacted contact mechanics and GH stability through the bone block effect. The main findings of this study were that the contribution of the bone block effect to GH stability increased with bone graft lateralization; however, as bone graft lateralization increased, so did peak contact pressures at the humeral head cartilage, which raises concerns for the increased risk of GH osteoarthritis. For the modeled conditions and the shoulder anatomy under analysis, the optimal bone graft placement balancing stability and peak contact pressure lay between a lateralization of 1.5 mm and 3.0 mm with respect to the flush position.

The positioning of the bone graft flush to the glenoid surface, as recommended in the literature [8] and named here as the ML-0 mm position, showed the bone block effect did not contribute to the anterior GH stability. For the medial–lateral positions considered in this study, anterior GH stability increased only after a lateralization of 3 mm with respect to the ML-0 mm position. However, the lateralization of the bone graft was also associated with a concerning increase in peak contact pressures caused by the incongruent interaction between the surfaces of the humeral head cartilage and the bone graft. Through experimental tests performed on 12 freshly frozen cadavers in static positions of humeral abduction, Ghodadra et al. [19] also observed an increase in articular contact pressure when bone grafts were placed 2 mm laterally with respect to a flush position. In addition to showing high peak contact pressures, the most lateralized position considered here, the ML-4.5 mm position, showed a noteworthy change in contact mechanics. During the compression of the humeral head into the glenoid cavity, the humeral head made contact with the bone graft, which caused its posterior translation. Although a comprehensive validation of the obtained contact pressures is not possible due to the lack of data, the fact that the peak contact pressures obtained from the GH joint model with a 20% glenoid bone defect before the anterior translation of the humeral head are within the range of peak contact pressures reported in the literature raises confidence in the results [33]. While considering cadaveric shoulders tested under a 440 N compressive force, Yamamoto et al. [33] found peak contact pressures in the 20% defected glenoid cartilage of 2.40 ± 0.46 MPa, 2.16 ± 0.49 MPa, and 3.70 ± 0.76 MPa for a shoulder in 30° abduction and in neutral rotation, a shoulder in 60° abduction and in neutral rotation, and a shoulder in 60° abduction with 90° of external rotation, respectively. In this study, peak contact pressures in the glenoid cartilage reached 1.21 MPa and 1.81 MPa for the shoulder in 0° abduction in the scapular plane and in neutral rotation, and 2.20 MPa and 3.08 MPa for the shoulder in 60° abduction in the scapular plane with 45° of external rotation under 50 N and 100 N compressive forces, respectively.

If a compromise between anterior GH stability and physiological contact pressures is sought, our results suggest that the optimal placement for the bone graft, as far as the contribution of the bone block effect is concerned, lay between a lateralization of 1.5 mm and 3.0 mm with respect to the ML-0 mm position, i.e., between a distance of −1.8 mm and 0.4 mm according to the axial circle method. However, the best position is likely to depend on subject-specific characteristics, such as bone graft geometry. Our results show that peak contact pressures are sensitive to millimetric variations in medial–lateral graft positioning, which suggests that, in practice, the ideal bone graft placement may be difficult to achieve in vivo. To document the accuracy of the bone–block position, Kany et al. [18] assumed an accurate position to be any position in which the graft was medialized by less than 5 mm or lateralized by less than 1 mm, which seems to acknowledge the difficulty and uncertainty associated with bone graft placement. For the bone block effect to contribute to GH stability, the bone graft must overhang from the glenoid surface at the cost of increasing the risk of osteoarthritis [6,7]. One possible solution to minimize the risk of osteoarthritis would be to consider alternative bone grafts that better mimic the glenoid cartilage geometry or to mold the coracoid bone graft in accordance with the glenoid cartilage to increase their congruity and smoothen their contact. Positive results have been found with bone grafts of the iliac crest [34], distal tibia [35], femoral condyle [36], or distal clavicle [37], but further investigation is necessary, as definitive evidence of their superiority is lacking [38].

The efficacy of the Latarjet procedure has been associated with three stabilizing mechanisms: the graft acting as a bone block; the conjoint tendon acting as a sling; and the lowering of the subscapularis muscle working as a hammock [27,39]. Using a custom testing machine to measure the anterior translational force necessary to dislocate cadaveric GH joints repaired by the Latarjet procedure, Yamamoto et al. [27] concluded that the sling effect was the main mechanism contributing to GH stability. Under this assumption, a conservative approach for the medial–lateral placement of the bone graft might be advisable. However, Moroder et al. [34] found similar clinical outcomes between coracoid and iliac crest graft transfers at the 6-, 12-, and 24-month follow-ups, which challenges the key role attributed to the sling effect of the Latarjet procedure. Further investigation is critical to understand the relevance of the stabilizing mechanisms of the Latarjet procedure and the suitability of alternative bone grafts in the long term.

To validate the finite element models developed, GH stability was compared with in vitro measurements for different conditions. Overall, the results obtained compare well to the literature [17,31], which provides confidence in the methodology and findings of this study. Nonetheless, this study is not without limitations. Similar to previous biomechanical studies [17,19], one of the most noteworthy limitations of this study is that the muscles and ligaments that contribute to GH stability were not modeled. Hence, from the three stabilizing mechanisms of the Latarjet procedure, only the bone block effect was studied. The modeling of the conjoint tendon, and the consequent simulation of the sling effect is likely to have different implications on GH stability ratios depending on arm elevation. Although recurrent anterior GH instability is often associated with both glenoid and humeral bone defects [40], only an isolated 20% glenoid bone defect was evaluated here. While investigating the impact of isolated and combined glenoid and humeral bone defects on GH stability, Walia et al. [40] verified that glenoid bone defects had a more significant impact on GH stability. Graft remodeling following the Latarjet procedure has been reported by several studies [41], but this phenomenon was not simulated here. Using suture-button fixation, instead of screw fixation, for the Latarjet procedure, Boileau et al. [42] and Xu et al. [43] reported a remodeling process that contributed to a glenoid curvature closer to its native geometry, which may improve the contact mechanics. For the sake of comparison, the loading conditions that were considered were similar to those of previous biomechanical studies, which do not correspond to true physiological loading conditions. Another limitation of this study is that only one geometry of the right shoulder was considered. As the best graft placement likely depends on shoulder geometry, the positioning between -1.8 mm and 0.4 mm, according to the axial circle method, deemed to be a compromise between stability gained by the bone block effect and the risk of osteoarthritis, might not apply to all patients. Finally, only a 20% anterior glenoid bone defect was studied, and variability in defect size and bone graft shape may affect shoulder stability and cartilage contact pressures.

## 5. Conclusions

According to the 3D finite element analyses performed, changing the medial–lateral positioning of the bone graft affects the contribution of the bone block effect to glenohumeral stability. However, while increased graft lateralization increases anterior glenohumeral stability, the overhang of the bone graft increases peak contact pressures at the humeral head cartilage, which raises concerns for the increased risk of glenohumeral osteoarthritis. For the modeled conditions and the shoulder anatomy under analysis, the optimal placement for the bone graft that balanced shoulder stability and the peak contact pressure lay between a lateralization of 1.5 mm and 3.0 mm with respect to the flush position.

## Figures and Tables

**Figure 1 biology-11-01783-f001:**
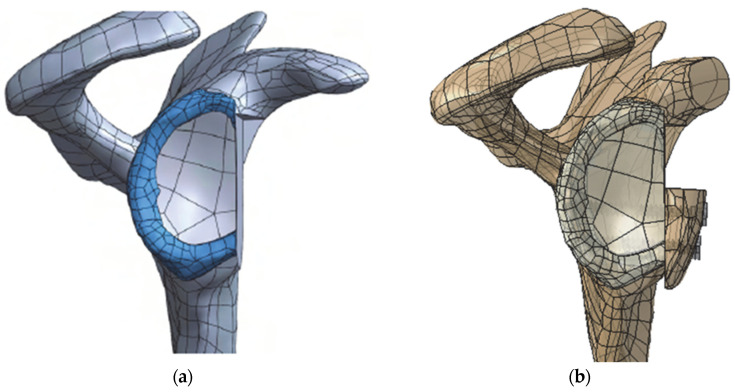
3D geometry of the scapula illustrating (**a**) the 20% glenoid bone defect simulated and (**b**) the Latarjet procedure with the bone graft in the reference position.

**Figure 2 biology-11-01783-f002:**
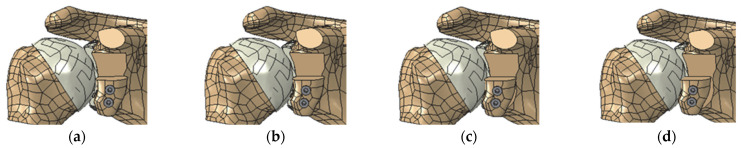
Anterior–posterior view of the medial–lateral positions considered for the bone graft in the Latarjet procedure: (**a**) ML-0 mm; (**b**) ML-1.5 mm; (**c**) ML-3.0 mm; and (**d**) ML-4.5 mm.

**Figure 3 biology-11-01783-f003:**
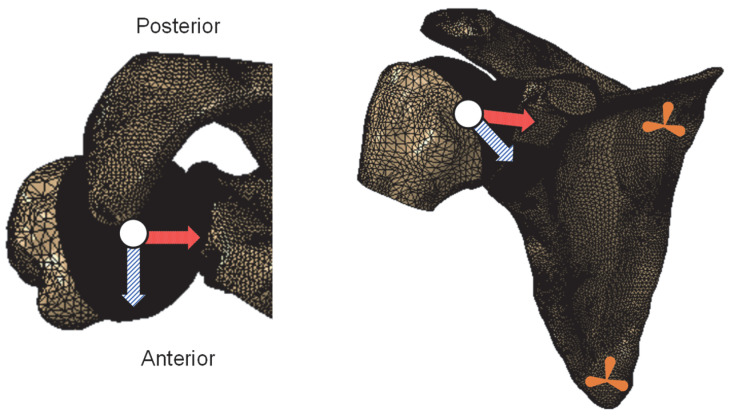
Illustration of the finite element meshes and loading and boundary conditions considered. Orange symbols on the scapula represent fully constrained displacement conditions. The red dotted arrows represent the direction of compression, while the arrows with diagonal stripes represent the anterior direction.

**Figure 4 biology-11-01783-f004:**
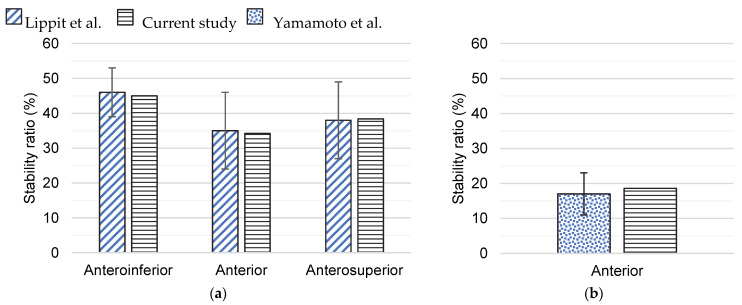
Stability ratios for (**a**) the healthy glenohumeral joint model and (**b**) the glenohumeral joint model with a 20% glenoid bone defect. For the sake of comparison, means and standard deviations of the experimental data of Lippit et al. [31] and Yamamoto et al. [17] are also presented.

**Figure 5 biology-11-01783-f005:**
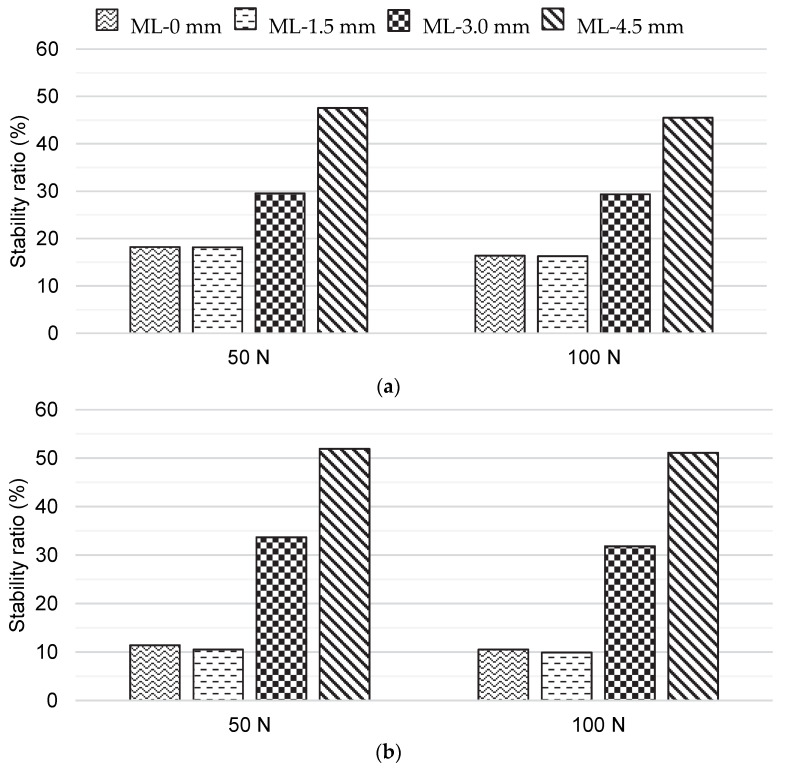
Stability ratios of the glenohumeral joint with a 20% glenoid bone defect after the Latarjet procedure for the medial–lateral graft positions modeled: (**a**) shoulder in 0° abduction in the scapular plane and in neutral rotation, and (**b**) shoulder in 60° abduction in the scapular plane with 45° of external rotation.

**Figure 6 biology-11-01783-f006:**
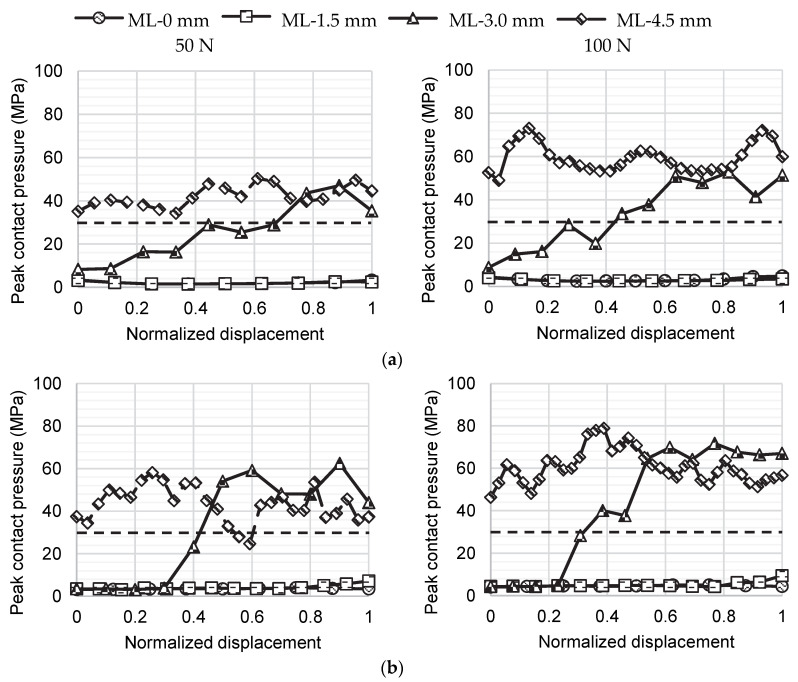
Evolution of peak contact pressure with anterior translation of the humeral head after the Latarjet procedure for the medial–lateral graft positions modeled: (**a**) shoulder in 0° abduction in the scapular plane and in neutral rotation and (**b**) shoulder in 60° abduction in the scapular plane with 45° of external rotation. The x-axis presents a normalized anterior translation of the humeral head in which 0 corresponds to the beginning of the humeral head translation and 1 corresponds to the position in which peak translation force was reached. The horizontal dashed line represents the failure stress (29.5 MPa) assumed for cartilage [32].

**Table 1 biology-11-01783-t001:** Evaluation of graft positions, in mm, according to the axial circle method. Positive distances mean the bone graft is lateralized, whereas negative distances mean the bone graft is medialized.

Latarjet Model	Distance (mm)
ML-0 mm	−3.2
ML-1.5 mm	−1.8
ML-3.0 mm	+0.4
ML-4.5 mm	+1.2

## Data Availability

Not applicable.

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
