# Peer review of "Influence of Graft Positioning during the Latarjet Procedure on Shoulder Stability and Articular Contact Pressure: Computational Analysis of the Bone Block Effect"

_biology, 2022, doi:10.3390/biology11121783_

Round 1
Reviewer 1 Report
The paper aims to clarify the efficacy of latarjet procedure in shoulder stability. The current literature is limited and divided about the effectiveness of lateralization. Using a mechanical (finite element) simulation, this paper clarifies that small lateralizations are not effective, yet about 3 mm. lateralization is most efficacious and further lateralization can lead to cartillage damage due to increased contact pressure. The model disregards (as the authors state) the added stability of tendons, still it clearly shows that the method is effective.
The geometric model was developed in SolidWorks. The model simulation was set up in ABAQUS software, using tetrahedral C3D4 and C3D4H finite elements. A nonhomogenous material model is used for bones (via bonemapy extension of ABAQUS) and a hyperelastic material model was selected for cartilage.
The boundary conditions are discussed in the text.
The adequacy of mesh density was evaluated and the model was first validated agains available data in the literature.
The efficacy of the lateralization was evaluated by computing a stability ratio, which is a reasonable indicator of stability.
The results are shown in terms of figures displaying stability ratio and contact pressure.
The methodology is robust, and scientifically sound. The discussions are to the point.
However, the presentation can be enriched by adding a figure in Materials and Methods section showing the C3D4 and C3D4H elements. A figure indicating boundary conditions would also be very useful. Also, a few figures showing deformations and stresses in the model, when the humerus bone is displaced.
The paper is very readable and the usage of English language is correct.
The results are very significant for the audience, and I believe it will have a good impact in the literature.
Author Response
We want to express our appreciation to the Reviewers for their constructive evaluation of our work that allowed us to improve it. In what follows, we discuss the changes performed to our work. We hope that our revision clearly addresses the Reviewers’ recommendations.
Reviewer #1
Remark 1: However, the presentation can be enriched by adding a figure in Materials and Methods section showing the C3D4 and C3D4H elements. A figure indicating boundary conditions would also be very useful. Also, a few figures showing deformations and stresses in the model, when the humerus bone is displaced.
Reply: Thank you for your comments.
As suggested by the Reviewer, the finite element meshes and the boundary conditions are now illustrated in the revised manuscript. For the sake of briefness, they are illustrated in a single figure (Figure 3 of the revised manuscript). The finite element models developed contain more than 1500000 elements, and the mesh density is especially high for the cartilages and bone graft. Consequently, for these components, Figure 3 does not allow a proper assessment of these meshes, other than their higher mesh density compared with the other components.
The original submission of this study was supposed to include a supplementary document in which the contact pressures at the contacting surfaces are shown. By mistake, this document was not submitted, but it is now in the revision process. We apologize for the inconvenience.
Reviewer 2 Report
This paper presents a modeling study that investigates an important clinical problem, the influence of graft positioning in the Latarjet procedure. The stability ratio results of the FE model were compared to experimental results, and a good match was achieved. The study is well performed, the manuscript is written in a clear and concise manner, and I enjoyed reading it. However, I do think that there are some important weaknesses. Here are the main ones, and how they could be reconsidered in the manuscript:
- Stability ratio was validated against experiment, but the reported contact pressures were not validated. The values reach very high values at very low compressive forces, and one may question these results. While I realize that contact pressure distribution plots are intended to be submitted as supplementary data, I could not access these figures in the reviewer version. How confident are you with respect to pressure values? I would suggest comparing the reported values with published work to see if these values are at least in the ballpark.
- The authors do mention that the FE model represents one single patient (N=1, line 280). This is indeed a major weakness, as the results may change completely with a different joint congruency. A very flat glenoid would require much less lateralization, while a very congruent glenoid may require more. I would urge the authors to investigate more of these parameters, which can be done relatively easily on a model.
- Also, the authors do mention the influence of graft geometry on line 234, which is also highly variable from patient to patient. This and the previous point makes me think that the optimal graft position should be provided with much caution. Therefore, I recommend to add that this was for that single patient anatomy everytime the optimal graft lateralization is given in the manuscript.
- Lines 60-61 provide existing papers that already demonstrated the main conclusions of the current study. So what aspects are truly new in the current study hypothesis (lines 69-71)?
DETAILED COMMENTS, WITH THE LINE NUMBER AT THE BEGINNING:
47: most OFTEN dislocated joint
83: replace humeral head by humeral articulation surface?
94: Why were screws required ? Why not just bond the bones and neglect the screws?
114: not why these two positions, compared to other positions?
137: why frictionless?
143: What is meant by superior and inferior angle of the scapula? Do you mean border? Please be more specific
141: The applied compression forces are very low. In order to get a sense of how much the total force was, could you also provide the values of the joint resultant force?
147: Why was the mesh refined on von Mises stress when the outputs of the model are stability ratio and contact pressure? The authors need to run mesh convergence on contact pressure.
149: Mesh convergence was stopped when results changed by less than 1%, but how where to mesh steps defined? For example, was the element length reduced in half in each iteration?
152: Can you please specify at what compressive forces the Lippit and Yamamoto studies were performed?
156: Suggest to rewrite: «modeling conditions reproduced the experimental test»
Figure 3: how sensitive are the model results to uncertainty? How do the result change with, for example, small changes in bone alignment, shear direction, material properties etc?
181: The previous sentence states that the ML 3.5 also provides increase stability, in accordance with Figure 4. So not sure what is meant here.
206-208: how was the optimum calculated? Why these values? Please describe.
209-211: So the current study contradicts these clinical results ?
Line 217-218: Could this data set be used for the above mentioned comment on validation of your contact pressure values?
Fig 5: Extremely high contact pressure values were obtained at very low joint compression forces. So in vivo, catastrophic pressure values can be expected. Why do these procedures not fail dramatically then? Is that a contact pressure validation issue?
230-234: Again, how are these values chosen. Please develop.
Author Response
We want to express our appreciation to the Reviewers for their constructive evaluation of our work that allowed us to improve it. In what follows, we discuss the changes performed to our work. We hope that our revision clearly addresses the Reviewers’ recommendations.
Reviewer #2
Remark 1: Stability ratio was validated against experiment, but the reported contact pressures were not validated. The values reach very high values at very low compressive forces, and one may question these results. While I realize that contact pressure distribution plots are intended to be submitted as supplementary data, I could not access these figures in the reviewer version. How confident are you with respect to pressure values? I would suggest comparing the reported values with published work to see if these values are at least in the ballpark.
Reply: We apologize for not having submitted the supplementary material. It was not submitted by mistake.
Disregarding the ML-4.5mm position, for which the bone graft seems to be too lateralized, peak contact pressures are not large before the anterior translation of the humeral head, i.e., at the end of the compression stage. They only increase to high values when contact with the bone graft occurs, which is likely due to the incongruent interaction between the surfaces of the humeral head cartilage and of the bone graft. For the shoulder in 0º of abduction in the scapular plane and neutral rotation, peak contact pressures at the glenoid cartilage reached 1.21 MPa and 1.81 MPa under 50 N and 100 N compressive forces, respectively, in the ML-0mm position. For the shoulder in 60º of abduction in the scapular plane and 45º external rotation, they reached 2.20 MPa and 3.08 MPa. Despite a comprehensive validation of results not being possible due to limited available data, these contact pressures are, nonetheless, within the range of values reported in the literature [1], which provides confidence in the results. Considering cadaveric shoulders tested under a 440 N compressive force, Yamamoto et al. [1] found mean peak contact pressures at the glenoid of 2.40 ± 0.46 MPa, 2.16 ± 0.49 MPa, and 3.70 ± 0.76 MPa, for the shoulder in 30º abduction and neutral rotation, the shoulder in 60º abduction and neutral rotation, and the shoulder in 60º abduction and 90º external rotation, respectively, considering a 20% bone defect. To address this, and provide further confidence in the results, the following text was added to the second paragraph of the discussion in the revised manuscript:
“Although a comprehensive validation of the obtained contact pressures is not possible due to the lack of data, the fact that peak contact pressures obtained with the GH joint model with a 20% glenoid bone defect before the anterior translation of the humeral head are within the range of peak contact pressures reported in the literature raises confidence in the results [33]. Considering cadaveric shoulders tested under a 440 N compressive force, Yamamoto et al. [33] found peak contact pressures at the 20% de-fected glenoid cartilage of 2.40 ± 0.46 MPa, 2.16 ± 0.49 MPa, and 3.70 ± 0.76 MPa, for the shoulder in 30º abduction and neutral rotation, the shoulder in 60º abduction and neutral rotation, and the shoulder in 60º abduction and 90º external rotation, respectively. In this study, peak contact pressures at the glenoid cartilage reached 1.21 MPa and 1.81 MPa for the shoulder in 0º of abduction in the scapular plane and neutral rotation, and 2.20 MPa and 3.08 MPa for the shoulder in 60º of abduction in the scapular plane and 45º external rotation, under 50 N and 100 N compressive forces, respectively.” (Lines 233-246 of the revised manuscript with Simple Markup)
Remark 2: The authors do mention that the FE model represents one single patient (N=1, line 280). This is indeed a major weakness, as the results may change completely with a different joint congruency. A very flat glenoid would require much less lateralization, while a very congruent glenoid may require more. I would urge the authors to investigate more of these parameters, which can be done relatively easily on a model.
Reply: We agree with the Reviewer that this is an important parameter that must be studied in further detail, and it is in our plans to study it in the near future. Nonetheless, we believe that the present study provides valuable insight into the contribution of the bone block effect to shoulder stability. Regarding this limitation, we tried to discuss it in the manuscript to raise the readers’ awareness to the conditions in which the findings of the study are based on.
Remark 3: Also, the authors do mention the influence of graft geometry on line 234, which is also highly variable from patient to patient. This and the previous point makes me think that the optimal graft position should be provided with much caution. Therefore, I recommend to add that this was for that single patient anatomy everytime the optimal graft lateralization is given in the manuscript.
Reply: We agree with the Reviewer and have revised the manuscript accordingly. For instance, the first paragraph of the discussion section was revised to:
“The main findings of this study were that the contribution of the bone block effect to GH stability increased with bone graft lateralization; however, as bone graft lateralization increased, so did peak contact pressures at the humeral head cartilage, which raises concerns of increased risk of GH osteoarthritis. For the modeled conditions and the shoulder anatomy under analysis, the optimal bone graft placement balancing stability and peak contact pressure lied between a lateralization of 1.5 mm and 3.0 mm with respect to the flush position.” (Lines 207-213 of the revised manuscript with Simple Markup)
Remark 4: Lines 60-61 provide existing papers that already demonstrated the main conclusions of the current study. So what aspects are truly new in the current study hypothesis (lines 69-71)?
Reply: The studies referred in the second paragraph of the introduction are retrospective studies that evaluated the outcome of the Latarjet procedure for different conditions, including the positioning of the bone graft. Bone graft positions were not quantified individually, but were categorized mainly into medial, flush, and lateral positions. In this study, a more objective range of graft positions is studied, providing insight into not only shoulder stability, but also peak contact pressure. The evolution of peak contact pressure with anterior translation is also presented, which cannot be measured in vivo.
Remark 5: 47: most OFTEN dislocated joint.
Reply: The manuscript was revised as suggested.
Remark 6: 83: replace humeral head by humeral articulation surface?
Reply: The manuscript was revised as suggested.
Remark 7: 94: Why were screws required ? Why not just bond the bones and neglect the screws?
Reply: For the current study, screws could have been neglected, as the Reviewer mentions. We modelled them because we intend to use the developed finite element model for other future studies as well.
Remark 8: 114: not why these two positions, compared to other positions?
Reply: We considered two shoulder positions because we wanted to evaluate how the biomechanics of the Latarjet procedure changed with shoulder motion. The shoulder positions considered were based on the positions studied in previous biomechanical studies [2–4]. To clarify this, the first sentence of the fourth paragraph of Section 2.1was revised to:
“All graft positions were evaluated for two shoulder positions, selected based on previous biomechanical studies [17,19,20]: a 0º abduction in the scapular plane and neutral rotation and a 60º abduction in the scapular plane and 45º external rotation.” (Lines 113-115 of the revised manuscript with Simple Markup)
Remark 9: 137: why frictionless?
Reply: Due to the lack of information regarding the in vivo friction conditions, we assumed a frictionless contact between articulating surfaces, as considered in previous studies (e.g. [5]).
Remark 10: 143: What is meant by superior and inferior angle of the scapula? Do you mean border? Please be more specific.
Reply: The regions of the scapula fixed were those, approximately, in the vicinity of the trigonum spinae and angulus inferior bony landmarks. To make it clearer in the manuscript, the referred text was revised to:
“The rotations of the humerus were constrained to keep the same arm position throughout the analysis and nodes in the vicinity of the trigonum spinae and angulus inferior bony landmarks of the scapula were fixed in all directions to avoid rigid body motion [28], as illustrated in Figure 3.” (Lines 140-144 of the revised manuscript with Simple Markup)
Figure 3 of the revised manuscript illustrates now the boundary conditions, which should help understanding where the scapula was fixed.
Remark 11: 141: The applied compression forces are very low. In order to get a sense of how much the total force was, could you also provide the values of the joint resultant force?
Reply: The joint resultant force is the sum of the compressive and tangential forces. As the stability ratio was always smaller than 60%, the joint resultant force was always smaller than 120 N. As stated in the manuscript, loading conditions considered were similar to those of previous biomechanical studies [6,7], even though they do not correspond to true physiological loading conditions. Nonetheless, given the comparative nature of the analyses performed in this study, we believe this limitation not to be critical.
Remark 12: 147: Why was the mesh refined on von Mises stress when the outputs of the model are stability ratio and contact pressure? The authors need to run mesh convergence on contact pressure.
Reply: We agree with the Reviewer. Actually, we first performed a sensitivity analysis based on peak translational force and Von Mises stress, but later evaluated, though less formally, the impact of the mesh size on the contact pressure. After this evaluation, we ended up selecting a mesh density that was far larger than that pointed by the initial sensitivity analysis (based on peak translational force and Von Mises stress). Note that the total number of elements and nodes resulting from the selected mesh density ranged between 1775000 and 1840000, and 350000 and 390000, respectively, in all Latarjet finite element models, which is quite large.
Following the Reviewer’s comment, we performed a (formal) sensitivity analysis on contact pressure considering the glenohumeral joint model with a 20% glenoid bone defect, in which average element size was changed from 0.8 mm to 0.2 mm with 0.1 mm increments, which confirmed that the mesh size initially selected was appropriate for the evaluation of the contact pressures. The largest difference between 2 consecutive iterations for the selected mesh size was not as low as 1%, but it was not much larger (1.3%) and was still much smaller than the 5% tolerance often considered in the literature [8,9]. To clarify the definition of the finite element meshes, the last paragraph of Section 2.2 was revised to:
“Mesh density was defined based on sensitivity analyses in which peak translational force was evaluated for the healthy glenohumeral joint model and peak contact pressure was evaluated for the glenohumeral joint model with a 20% glenoid bone defect. Mesh refinement, defined from average element sizes of 0.8 mm to 0.2 mm, with 0.1 mm increments, was stopped when results changed less than 5% in two consecutive iterations [29,30]. The total number of elements and nodes resulting from the selected mesh density ranged between 1775000 and 1840000, and 350000 and 390000, respectively, in all Latarjet finite element models.” (Lines 150-157 of the revised manuscript with Simple Markup)
Remark 13: 149: Mesh convergence was stopped when results changed by less than 1%, but how where to mesh steps defined? For example, was the element length reduced in half in each iteration?
Reply: Meshes considered during the convergence analyses varied 0.1 mm in the average element size parameter between consecutive iterations. This information is now provided in the last paragraph of Section 2.2 of the revised manuscript:
“Mesh density was defined based on sensitivity analyses in which peak translational force was evaluated for the healthy glenohumeral joint model and peak contact pressure was evaluated for the glenohumeral joint model with a 20% glenoid bone defect. Mesh refinement, defined from average element sizes of 0.8 mm to 0.2 mm, with 0.1 mm increments, was stopped when results changed less than 5% in two consecutive iterations [29,30]. The total number of elements and nodes resulting from the selected mesh density ranged between 1775000 and 1840000, and 350000 and 390000, respectively, in all Latarjet finite element models.” (Lines 150-157 of the revised manuscript with Simple Markup)
Remark 14: 152: Can you please specify at what compressive forces the Lippit and Yamamoto studies were performed?
Reply: For the validation simulations we considered a 50 N compressive force, considered both by Lippit et al. [10] and Yamamoto et al. [2]. To better clarify this in the manuscript, the first paragraph of Section 2.3 was revised to:
“Following the study of Lippit et al. [31], the humeral head in the healthy GH joint model was compressed onto the glenoid by a 50 N force and was translated in the anteroinferior, anterior, and anterosuperior directions.” (Lines 162-165 of the revised manuscript with Simple Markup)
Remark 15: 156: Suggest to rewrite: «modeling conditions reproduced the experimental test».
Reply: The manuscript was revised as suggested.
Remark 16: Figure 3: how sensitive are the model results to uncertainty? How do the result change with, for example, small changes in bone alignment, shear direction, material properties etc?
Reply: To address the Reviewer’s concern, sensitivity analyses on the bone translation direction and cartilage material properties were performed. The sensitivity of the stability ratios to these modelling parameters was found to be limited. Increasing the C10 parameter of the Neo-Hookean material considered for cartilage by 25%, while keeping the remaining parameters constant, changed the stability ratio from 18.6% to 19.18%. The decreasing of the same parameter by 25%, decreased the stability ratio to 17.87%. Regarding the bone translation direction, rotations of +25º and -25º degrees were applied to the anterior translation direction considered in the validation procedure described in the manuscript. The stability ratios obtained were 17.38% and 22.88% for the positive and negative rotations, respectively. All these stability ratios remained within 1 standard deviation of the experimental results presented by Yamamoto et al. [2].
Remark 17: 181: The previous sentence states that the ML 3.5 also provides increase stability, in accordance with Figure 4. So not sure what is meant here.
Reply: In the referred lines, we highlight two main results: GH stability increased only for the ML-3.0mm and ML-4.5mm positions; and, for the ML-4.5mm position, GH stability became even larger than in the healthy condition—the stability ratio in the anterior direction was 47.56% for the ML-4.5mm position, while it was only 34.20% for the healthy condition. To make these two main results clearer, the text in the referred lines was revised to:
“The bone block effect of the Latarjet procedure contributed to GH stability only in the ML-3.0mm and ML-4.5mm positions, regardless of the shoulder position, as shown in Figure 5. For the ML-4.5mm position, GH stability was greater than in the healthy GH joint.” (Lines 180-183 of the revised manuscript with Simple Markup)
Remark 18: 206-208: how was the optimum calculated? Why these values? Please describe.
Reply: Assuming that the optimal placement for the bone graft must balance anterior GH stability and physiological contact pressures, to decrease both the risk of instability recurrence and of osteoarthritis, our results suggest that, as far as the bone block effect is concerned, the bone graft should be placed between a lateralization of 1.5 mm and 3.0 mm with respect to the ML-0mm position. To better clarify this in the manuscript, the last sentence of the first paragraph of the discussion section was revised to:
“For the modeled conditions and the shoulder anatomy under analysis, the optimal bone graft placement balancing stability and peak contact pressure lied between a lateralization of 1.5 mm and 3.0 mm with respect to the flush position.” (Lines 210-213 of the revised manuscript with Simple Markup)
Remark 19: 209-211: So the current study contradicts these clinical results ?
Reply: The results from our study do not contradict clinical results. Clinical studies evaluate the Latarjet procedure as a whole, considering all stabilizing mechanisms associated with the Latarjet procedure, while our study focuses only on the bone block effect of the Latarjet procedure. Consequently, its findings must be analyzed bearing this in mind, as often highlighted in the manuscript to ensure no wrong message is taken from our study.
Remark 20: Line 217-218: Could this data set be used for the above mentioned comment on validation of your contact pressure values?
Reply: The data reported by Ghodadra et al. [3] could be used for the validation of the contact pressures, but we ended up using the data from Yamamoto et al. [1] for that purpose, as detailed in our reply to Remark 1. The reason for this decision was that the results from Yamamoto et al. [1] include data for a glenoid bone defect of 20%, whereas those from Ghodadra et al. [3] do not. The study by Ghodadra et al. [3] considers glenoid bone defects of 15% and 30%. Note, nonetheless, that our results are coherent with theirs. In the study by Ghodadra et al. [3], peak contact pressures for the 15% and 30% glenoid bone defects ranged between 2.30 MPa and 6.35 MPa under a compressive force of 440 N, while in our study, they ranged between 1.21 MPa and 3.08 MPa under compressive forces of 50 N and 100 N.
Remark 21: Fig 5: Extremely high contact pressure values were obtained at very low joint compression forces. So in vivo, catastrophic pressure values can be expected. Why do these procedures not fail dramatically then? Is that a contact pressure validation issue?
Reply: Extremely high contact pressures were only obtained when contact with the bone graft occurred. According to the literature, the bone graft is recommended to be placed flush or slightly medial to the glenoid surface [5], which, based on our results, may prevent the appearance of high contact pressures. One possible reason for the Latarjet procedure not failing “dramatically” due to high contact pressures may be the fact that most surgeons do not place the bone graft in a lateralized position. For instance, in the study by Mizuno et al. [11] only 13% out of 68 patients had lateral overhang of the bone graft, defined as the lateral aspect of the bone graft protruding 1 mm beyond the glenoid surface. In the group of patients with postoperative osteoarthritis (OA) or progression of OA, 43.7% of patients had lateral overhang of the bone graft, whereas in the group without postoperative OA or progression of OA, only 3.8% had lateral overhang of the graft. Additionally, in the long-term, a remodeling process of the bone graft that contributes to a glenoid curvature closer to its native geometry, as reported by Boileau et al. [12] and Xu et al. [13], may help improve contact mechanics and prevent a catastrophic failure of the Latarjet procedure.
Remark 22: 230-234: Again, how are these values chosen. Please develop.
Reply: The bone graft positioning between a lateralization of 1.5 mm (ML-1.5mm) and 3.0 mm (ML-3.0mm) with respect to the flush position is selected assuming a compromise between anterior GH stability and physiological contact pressures. For the ML-1.5 mm position, the humeral head never contacted the bone graft, which means that the bone block effect did not contribute to shoulder stability. Note, however, that this does not mean that the Latarjet procedure would not provide stability, as its other stabilizing mechanisms, not modelled in this study, could still be effective. For the ML-3.0mm, the humeral head contacted the bone graft, but the increase in stability was associated with an increase in peak contact pressure. Based on this trade-off between stability and the risk of osteoarthritis, the optimal placement for the bone graft considering the bone block effect alone seems to lie between a lateralization of 1.5 mm and 3.0 mm with respect to the flush position.
References
[1] Yamamoto A, Massimini DF, Distefano J, Higgins LD. Glenohumeral contact pressure with simulated anterior labral and osseous defects in cadaveric shoulders before and after soft tissue repair. Am J Sports Med 2014;42:1947–54. https://doi.org/10.1177/0363546514531905.
[2] Yamamoto N, Itoi E, Abe H, Kikuchi K, Seki N, Minagawa H, et al. Effect of an anterior glenoid defect on anterior shoulder stability: A cadaveric study. Am J Sports Med 2009;37:949–54. https://doi.org/10.1177/0363546508330139.
[3] Ghodadra N, Gupta A, Romeo AA, Bach BR, Verma N, Shewman E, et al. Normalization of glenohumeral articular contact pressures after Latarjet or iliac crest bone-grafting. J Bone Jt Surg - Ser A 2010;92:1478–89. https://doi.org/10.2106/JBJS.I.00220.
[4] Boons HW, Giles JW, Elkinson I, Johnson JA, Athwal GS. Classic versus congruent coracoid positioning during the latarjet procedure: An in vitro biomechanical comparison. Arthrosc - J Arthrosc Relat Surg 2013;29:309–16. https://doi.org/10.1016/j.arthro.2012.09.007.
[5] Sano H, Komatsuda T, Abe H, Ozawa H, Yokobori TA. Proximal-medial part in the coracoid graft demonstrates the most evident stress shielding following the Latarjet procedure: a simulation study using the 3-dimensional finite element method. J Shoulder Elb Surg 2020;29:2632–9. https://doi.org/10.1016/j.jse.2020.03.037.
[6] Walia P, Miniaci A, Jones MH, Fening SD. Theoretical model of the effect of combined glenohumeral bone defects on anterior shoulder instability: A finite element approach. J Orthop Res 2013;31:601–7. https://doi.org/10.1002/jor.22267.
[7] Yamamoto N, Muraki T, Sperling JW, Steinmann SP, Cofield RH, Itoi E, et al. Stabilizing mechanism in bone-grafting of a large glenoid defect. J Bone Jt Surg - Ser A 2010;92:2059–66. https://doi.org/10.2106/JBJS.I.00261.
[8] Lee HK, Kim SM, Lim HS. Computational Wear Prediction of TKR with Flatback Deformity during Gait. Appl Sci 2022, Vol 12, Page 3698 2022;12:3698. https://doi.org/10.3390/APP12073698.
[9] Erbulut DU, Sadeqi S, Summers R, Goel VK. Tibiofemoral Cartilage Contact Pressures in Athletes During Landing: A Dynamic Finite Element Study. J Biomech Eng 2021;143. https://doi.org/10.1115/1.4051231.
[10] Lippitt SB, Vanderhooft JE, Harris SL, Sidles JA, Harryman DT, Matsen FA. Glenohumeral stability from concavity-compression: A quantitative analysis. J Shoulder Elb Surg 1993;2:27–35. https://doi.org/10.1016/S1058-2746(09)80134-1.
[11] Mizuno N, Denard PJ, Raiss P, Melis B, Walch G. Long-term results of the Latarjet procedure for anterior instability of the shoulder. J Shoulder Elb Surg 2014;23:1691–9. https://doi.org/10.1016/j.jse.2014.02.015.
[12] Boileau P, Saliken D, Gendre P, Seeto BL, d’Ollonne T, Gonzalez JF, et al. Arthroscopic Latarjet: Suture-Button Fixation Is a Safe and Reliable Alternative to Screw Fixation. Arthrosc - J Arthrosc Relat Surg 2019;35:1050–61. https://doi.org/10.1016/j.arthro.2018.11.012.
[13] Xu J, Liu H, Lu W, Deng Z, Zhu W, Peng L, et al. Modified Arthroscopic Latarjet Procedure: Suture-Button Fixation Achieves Excellent Remodeling at 3-Year Follow-up. Am J Sports Med 2020;48:39–47. https://doi.org/10.1177/0363546519887959.
Round 2
Reviewer 2 Report
Contact pressure plots:
- In the current form, the value of the contact pressure distribution is very limited. Most images only display a blue image because the peak is a tenth of the maximum scale value. I suggest to chose a different scale, potentially a different one for each image, so that the contacts can be seen on all images.
- The distribution is not homogenous within the high contact pressure region. Is that due to ill-defined contacts, or meshing issue?
- Please add the units
Author Response
We want to express our appreciation to the Reviewer for the constructive evaluation of our work that allowed us to improve it. We hope that our revision, described in detail next, clearly addresses the Reviewer’s recommendations.
Reviewer #2
Remark 1: In the current form, the value of the contact pressure distribution is very limited. Most images only display a blue image because the peak is a tenth of the maximum scale value. I suggest to chose a different scale, potentially a different one for each image, so that the contacts can be seen on all images.
Reply: As suggested by the Reviewer, we added new figures to the supplementary material document considering different scales for each bone graft position. To continue to allow a more direct comparison of results, we also maintained the original figures, with the same scale for all bone graft positions, in the document.
Remark 2: The distribution is not homogenous within the high contact pressure region. Is that due to ill-defined contacts, or meshing issue?
Reply: For some finite element models, contact pressure distributions presented some noise within high contact pressure regions likely due to the type of finite element used. While comparing the advantages and disadvantages of tetrahedral and hexahedral elements, Tadepalli et al. [1] found a similar behavior in linear tetrahedral elements (the elements used in our study) when evaluating contact pressure, which was not observed in linear hexahedral or quadratic tetrahedral elements.
Remark 3: Please add the units.
Reply: As suggested by the Reviewer, the units were added to the captions of all figures in the supplementary material document.
References
[1] Tadepalli SC, Erdemir A, Cavanagh PR. Comparison of hexahedral and tetrahedral elements in finite element analysis of the foot and footwear. J Biomech 2011;44:2337–43. https://doi.org/10.1016/j.jbiomech.2011.05.006.